# Genome-Wide Analysis of Genes Involved in the GA Signal Transduction Pathway in ‘*duli*’ Pear (*Pyrus betulifolia* Bunge)

**DOI:** 10.3390/ijms23126570

**Published:** 2022-06-12

**Authors:** Pingli Song, Gang Li, Jianfeng Xu, Qingcui Ma, Baoxiu Qi, Yuxing Zhang

**Affiliations:** 1College of Horticulture, Hebei Agricultural University, Baoding 071000, China; pinglisong2022@163.com (P.S.); liganghebau@163.com (G.L.); xjf@hebau.edu.cn (J.X.); maqingcui2020@163.com (Q.M.); 2School of Pharmacy and Biomolecular Sciences, Liverpool John Moores University, Liverpool L3 3AF, UK

**Keywords:** ‘*duli*’ pear, *Pyrus betulifolia* Bunge, GA signaling, GID1, DELLA, SLY

## Abstract

Gibberellic acid (GA) is an important phytohormone that regulates every aspect of plant growth and development. While elements involved in GA signaling have been identified and, hence, their functions have been well studied in model plants, such as Arabidopsis and rice, very little is known in pear. We, therefore, analyzed the genes related to GA signaling from the recently sequenced genome of the wildtype ‘*duli*’ pear (*Pyrus betulifolia* Bunge), a widely used rootstock for grafting in pear cultivation in China due to its vigorous growth and resistance to abiotic and biotic stress. In total, 15 genes were identified, including five GA receptors *PbGID1s* (*GA-INSENSTIVE DWARF 1*), six GA negative regulators, *PbDELLAs*, and four GA positive regulators, *PbSLYs*. Exogenous application of GA could promote the expression of *PbGID1s* but inhibit that of *PbDELLAs* and *PbSLYs* in tissue culture ‘*duli*’ pear seedlings. The expression profiles of these genes in field-grown trees under normal growth conditions, as well as in tissue-cultured seedlings treated with auxin (IAA), GA, paclobutrazol (PAC), abscisic acid (ABA), and sodium chloride (NaCl), were also studied, providing further evidence of the involvement of these genes in GA signaling in ‘*duli*’ pear plants. The preliminary results obtained in this report lay a good foundation for future research into GA signaling pathways in pear. Importantly, the identification and preliminary functional verification of these genes could guide molecular breeding in order to obtain the highly desired dwarf pear rootstocks for high-density plantation to aid easy orchard management and high yielding of pear fruits.

## 1. Introduction

Gibberellins or gibberellic acids (GAs), tetracyclic diterpenoid phytohormones, are widely distributed in higher plants [1,2]. They play an important role in seed germination, hypocotyl elongation, flowering time transition, seed and fruit development, and abiotic and biotic stress response [3,4,5,6]. Since the discoveries of the GA-insensitive dwarf mutants of wheat and rice, one of the most decisive players in ‘Green Revolution’ in the late 1960s which led to a massive increase in grain production, saving millions of lives in developing countries, much effort has been put in place to unravel the molecular mechanism of the GA signal transduction pathway in plants. As a result, the key genes involved in the GA signaling pathway have been isolated, including those encoding for GID1s, DELLAs, and SLYs from various plants [4,7,8].

GID1, a cytosolic gibberellin receptor, was first discovered in rice (*OsGID1*) for its ability to sense and bind active GAs, transmit GA signals, and induce downstream reactions [9]. Subsequently, many other GID1s were also successfully isolated and studied in other plant species [3,10], the first of which were the three *OsGID1* homologous genes from Arabidopsis, *AtGID1a*, *AtGID1b*, and *AtGID1c*. All three AtGID1s could strongly bind active GA and physically interact with DELLA proteins in the presence of GA [11]. They are also functionally redundant because the single mutant has no obvious GA-insensitive phenotype, whilst the triple mutant is dwarfed with a complete loss of response to GA [11,12,13,14]. However, a recent study showed that, while *AtGID1c* positively regulated seed germination, *AtGID1b* suppressed germination during seed dormancy in the dark, indicating that these two GID1s function independently during seed germination in Arabidopsis [15]. Furthermore, a recent study in tomato using CRISPR/Cas9-edited plants showed that, although the three tomato *GID1* genes are functionally redundant, as only the triple-knockout mutant, *gid1*^TRI^, showed the typical GA-insensitive phenotype, this redundancy disappeared under field conditions [16]; therefore, these three *GID1s* play overlapping, yet specific roles in maintaining the optimum growth and development of tomato plant [16]. However, in peach, the GA-insensitive dwarf mutant *‘shouxing’* is caused by the S191F mutation in PbGID1c, i.e., loss of function of a single GID1 can result in dwarfism [17]. Therefore, the combined results from these studies clearly demonstrated that GID1s in different plant species participate in different aspects of GA-mediated growth and development either independently or redundantly. 

The second GA signaling component, perhaps the most ‘famous’ one is the ‘Green Revolution’ gene, *DELLA*, first described in the 1960s but only cloned some 30 years later in the 1990s from wheat [18]. DELLAs, named after the highly conserved DELLA (Asp–Glu–Leu–Leu–Ala) amino-acid sequence in their N-termini, are negative regulators of GA signaling. They belong to the GRAS (named after the three DELLA proteins GAI, RGA, and SCR) protein family [19]. In addition to the DELLA domain, the N-termini contain the TYHYNP (Thr–Tyr–His–Tyr–Asn–Pro) domain, which plays an important role in the transduction of GA signal and maintains the stability of the DELLA protein. Other functional domains are the nuclear localization signal (NLS) and VHIID (Val–His–Ile–Ile–Asp) domain in the middle, and the LZ region (leucine repeat region), PFYRE (Pro–Phe–Tyr–Arg–Glu) domain, repressor domain SAW (Ser–Ala–Trp), and RVER (Arg–Val–Glu–Arg) domain in the C-termini [20]. 

*DELLAs* have been identified in Arabidopsis (*GAI*), rice (*Oryza sativa*, *SLR1*) [21], wheat (*Triticum aestivum*, *Rht-B1b* and *Rht-D1b*) [18], corn (*Zea mays*, *D8*) [18], barley (*Hordeum vulgare*, *SLN1*) [22], and grape (*Vitis vinifera*, *VvGAI*) [23]. Mutations/deletions in the DELLA domain can cause GA-insensitive and dominant or semi-dominant dwarf phenotypes in these plants, demonstrating that the integrity of the DELLA motif plays a pivotal role in the function of this family of proteins. For example, the GA-insensitive mutant *gai* of Arabidopsis has the N-terminal 17 amino acids which contain the DELLA domain of GAI missing. The mutant has a distinctive phenotype where dwarfism and small, dark leaves are among the most recognizable features, which are also found in the *gid1* triple mutant [19,24]. Four *GAI* homologous genes, *RGA*, *RGL1*, *RGL2*, and *RGL3*, were subsequently identified in Arabidopsis. They participate in seed germination and flower development (*RGA*, *RGL1*, and *RGL2*) and also play roles in stress responses (*RGL1, RGL2*, and *RGL3*) [25,26,27].

The third important component of GA signaling is the SCF (Skpl/cullin/F-box) of the E3 ubiquitin ligase. SCF plays essential roles in the ubiquitination and degradation of DELLA through the 26S proteasome system. As the subunit of SCF, the F-box protein determines the specificity for substrate DELLA recognition. Therefore, mutation in this protein can block GA signaling and, hence, cause GA insensitivity of the mutant [28]. For example, *SLY1* (*SLEEPY1*) and *SLY2/SNE* (*SLEEPY2/SNEEZY*) in Arabidopsis and their homologous gene *OsGID2* in rice encode for such F-box proteins [28,29]. The Arabidopsis mutant *sly1* is GA-insensitive due to its inability to degrade DELLA [30]. The fact that the double mutant *sly1sne* is more dwarfed and less fertile than the single mutant *sly1* indicates that both *SLY1* and *SNE* are functionally redundant in positively regulating the action of GA in growth [31,32].

The molecular mechanism of GA signaling in plant has become clearer in recent years owing to the genetic identification of many GA-related mutants in various plant species as mentioned above. It is believed that the active GA molecule first binds to the GA receptor GID1 to form a GA–GID1 complex which can bind to the DELLA domain of the DELLA protein. This results in a conformational change of DELLA. The formation of the GA–GID1–DELLA complex in turn allows the C-terminus of DELLA to interact with the SCF ^SLY1/GID2/SNE^ complex. This leads to the ubiquitination and subsequent degradation of DELLA by the proteasome complex, resulting in the release of GA from the GA–GID1–DELLA complex. GA can then enter the nucleus to activate the expression of downstream genes, thereby promoting GA-related growth and development [33].

While GA signaling in the regulation of architecture, fertility, and stress is well characterized in Arabidopsis, rice, and other crop plant species, this knowledge in woody plants, such as pear tree, is very limited. Given the important role of DELLAs and other GA signaling components in the regulation of plant height and the lack of dwarf pear rootstocks, we aimed to first identify *GID1*, *DELLA*, and *SLY* homologous genes from the wildtype ‘*duli*’ pear genome. Subsequently, the evolution and expression profiles of these genes in various tissues of 5 year old ‘*duli*’ pear trees under normal growth conditions, as well as in tissue-cultured seedlings treated with IAA, GA, PAC, ABA, and NaCl, were assessed in order to dissect their biological functions. The preliminary data obtained in this study could provide valuable information for further research of these genes, which could guide molecular breeding to create the much-desired dwarf ‘*duli*’ pear rootstocks in future.

## 2. Results

### 2.1. The Genome of ‘duli’ Pear Encodes 15 GA Signaling-Related Genes Located on 11 Different Chromosomes

Fifteen homologous genes related to GA signaling were identified from the ‘*duli*’ pear genome (Table 1). Specifically, there were five GID1-encoding genes (*PbGID1s*) distributed on four of the 17 chromosomes (Chr3.g17752.m1, Chr4.g40571.m1, and Chr11.g13248.m1 on the third, fourth, and 11th chromosomes; Chr12.g35366.m1 and Chr12.g35281.m1 on the 12th chromosome) (Figure 1, green). Their coding regions were between 1035 and 1095 bp, encoding proteins of 344–364 amino acids with molecular weights between 38.60 and 41.23 kDa. 

Six *PbDELLAs* were identified, distributed on chromosomes 2, 9 13, 15, 16, and 17 (Figure 1, blue). It was noted that the coding regions of these six *PbDELLAs* were highly variable in length (1614–2364 bp), number of amino acids (537–787), and molecular weight (58.8–86.3 kDa) of their encoded proteins. 

A homologous search also discovered four *SLY* genes (*PbSLYs*). They were much smaller than both *PbGID1s* and *PbDELLAs*, whereby the CDS of the largest one was only 720 bp long (Chr8.g54419.m1), located on chromosome 8. It encodes a putative protein of 239 amino acids and 26.3 kDa in molecular weight, while the other three *PbSLYs* were located on chromosomes 9, 15, and 17 (Table 1; Figure 1, red). 

### 2.2. Phylogenetic Analysis Identified Potential Dwarf-Related Genes from PbGID1s, PbDELLAs, and PbSLYs 

Phylogenetically related proteins usually share similar biological functions. Protein sequence alignment of GID1s showed that they fall into two types (Figure 2). Type A, including Chr12.g35366.m1, Chr12.g35281.m1, and Chr4.g40571.m1, had the closest sequence similarity to peach PpGID1c (XP_007207928), followed by Arabidopsis AtGID1a (AT3G05120) and AtGID1c (AT5G27320), and tomato LeGID1a (Solyc01g098390). Therefore, we named these three PbGID1s PbGID1c-1-1 (Chr12.g35366.m1), PbGID1c-1-2 (Chr12.g35281.m1), and PbGID1c-2 (Chr4.g40571.m1). Type B contained the remaining two members, Chr3.g17752.m1 and Chr11.g13248.m1; they were most closely related to AtGID1b (AT3G63010), LeGID1b1 (Solyc09g074270), LeGID1b2 (Solyc06g008870), and PpGID1b (XP_007200347). Hence, we named them PbGID1b-1 (Chr3.g17752.m1) and PbGID1b-2 (Chr11.g13248.m1). 

Similarly, we compared the protein sequences of the six putative PbDELLAs with known DELLAs from other plant species. This led to their classification into three groups. The first group with two ‘*duli*’ pear DELLAs, Chr16.g31263.m1 and Chr13.g24360.m1, also contained apple MdRGL1a (DQ007885) and MdRGL1b (DQ007886), Arabidopsis AtGAI (AT1G14920) and AtRGA (AT2G01570), grape VvGAI (AF378125), barley SLN1 (AF460219), wheat Rht-D1a (AJ242531), rice OsSLR1 (AB030956), and maize D8 (AJ242530). Importantly, this group of DELLAs is well known for their regulation of plant height because their knockout mutants exhibit dwarfism, such as wheat *rht-D1a1* and rice *sly1* [18,22,34,35,36,37]. Therefore, we named Chr16.g31263.m1 and Chr13.g24360.m1 as PbGAI1a and PbGAI1b, respectively. The second group included Chr9.g44536.m1 and Chr17.g27556.m1, which were named as PbGAI2a and PbGAI2b because their closest homologous proteins were MdRGL2s (DQ007883, DQ007884) from apple, which were also shown to affect plant height [38]. The last two members, Chr15.g03238.m1 and Chr2.g41461.m1 in the third group of DELLAs, shared the highest sequence similarity with apple MdRGL3a/b (DQ007887, DQ007888); hence, they were named as PbRGLa and PbRGLb, respectively (Figure 2). 

Lastly, we analyzed the sequences of the four PbSLYs by comparing them with those from Arabidopsis (AtSLY1, AT4G24210 and AtSLY2, AT5G487170) and rice (OsGID2, BAC81428). This resulted in their classification into two groups with each group containing two members. Accordingly, they were named as PbSLY1-1 (Chr8.g54419.m1) and PbSLY1-2 (Chr15.g04246.m1) in the group SLYⅠ, and PbSLY2-1 (Chr9.g46276.m1) and PbSLY2-2 (Chr17.g25753.m1) in the group SLYIⅠ (Figure 2).

Therefore, the ‘*duli*’ pear genome has a full repertoire of GA signaling components. Importantly, the potential dwarf-related genes are also present within these three families of genes.

### 2.3. PbGID1s, PbDELLAs, and PbSLYs Contain the Typical Conserved Domains and Motifs 

As shown in Figure 3a, all PbGID1s contained two conserved domains, the co-esterase (pfam00135) and α/β hydrolase-3 (pfam07859) belonging to the α/β hydrolase family. These PbGID1s also contained 10 conserved motifs (Appendix A) located in similar positions within the individual PbGID1s (Figure 3a).

Analysis of the six PbDELLAs showed that they all contained the characteristic conserved DELLA (pfam12041) domain in their N-termini and the GRAS (pfam03514) domain in the C-termini (Figure 3b). Note that the DELLA (motif 5) showed some variations, where the amino acids EL in PbRGLa and PbRGLb were replaced by GC and GY ((DGCLA and DGYLA instead of DELLA), respectively. It is also noteworthy that the sequence of PbGAI1b was different from the other five PbDELLAs in that it contained a putative transmembrane domain between amino acids 2 and 24, implying that it may have a different function (Figure 3b). With the exception of PbGAI2b that did not have motif 9 SAW close to the C-terminus, a typical repressor motif of the GRAS family proteins, all other PbDELLAs had the 10 conserved motifs (Appendix A). 

Conserved domains and motifs of PbSLYs were also analyzed, and the results are shown in Figure 3c. Only one conserved domain, the F-box (pfam00646) domain, was present. Conserved motifs 1, 2, 3, and 6 were found in the sequences of all four PbSLYs, whilst motif 4 and motif 5 were only found in PbSLY2s and PbSLY1s, respectively, suggesting that these two types of PbSLYs may have different roles in GA signaling.

### 2.4. Synteny and Duplication Analysis of PbGID1s, PbDELLAs, and PbSLYs of ‘duli’ Pear

Understanding the gene duplication events occurring in the genome and the synteny between different genomes can help understand gene evolution and function. Gene duplication, including tandem duplication, segmental duplication, and whole-genome duplication (WGD), is the major driving force in plant evolution. A multigene family is the result of region-specific duplication of the genome or WGD [39]. To evaluate the gene duplication events for *PbGID1s, PbDELLAs*, and *PbSLYs,* synteny and selective pressure analyses were carried out, and the results are presented in Figure 4a. *PbGID1c-1-1* had a syntenic relationship with *PbGID1c-1-2* and *PbGID1c-2*, as did *PbGID1b-1* with *PbGID1b-2*. For *PbDELLAs*, *PbGAI1a* and *PbGAI1b* had a syntenic relationship, as did *PbRGLa* and *PbRGLb*. For *PbSLYs,* a syntenic relationship was found between *PbSLY1-1* and *PbSLY1-2*, as well as between *PbSLY2-1* and *PbSLY2-2*. All these gene pairs were the result of WGD or segmental duplications. 

Next, the syntenic relationships between the genes of each family of *GID1s, DELLAs*, and *SLYs* were compared between ‘*duli*’ pear and Arabidopsis. As shown in Figure 4b, *PbGID1c-1-1*, *PbGID1c-1-2*, and *PbGID1c-2* had synteny with Arabidopsis *AtGID1c*, of which *PbGID1c-1-2* and *PbGID1c-2* also had synteny with *AtGID1a*. Both *PbGID1b-1* and *PbGID1b-2* were syntenic with *AtGID1b*. For the *PbDELLAs*, *PbGAI1a* and *PbGAI1b* had a syntenic relationship with *AtGAI* and *AtRGA*, respectively, as did *PbGAI2a* with *AtRGL1*. *PbSLY2-2* had a syntenic relationship with *AtSLY2/AtSNE*. 

Therefore, on the basis of the syntenic relationship between these genes in ‘*duli*’ pear and Arabidopsis, they are divided into two groups. The first group, including *PbGAI2b, PbRGLa*, and *PbRGLb* of the *PbDELLA* family and *PbSLY1-1*, *1-2*, and *2-1* of the *PbSLY* family, was due to ‘dispersed’ duplication, indicating a separation by other sequences where the genes may have arisen from transposition, such as ‘replicative transposition’, ‘non-replicative transposition’, or ‘conservative transposition’. The second group originated from WGD or segmental duplication, and it included all *PbGID1**s*, *PbGAI1a*, *PbGAI1b,* and *PbGAI2a* of the *PbDELLAs*, as well as *PbSLY2-2*.

Ka/Ks represents the ratio between the nonsynonymous substitution rate (Ka) and synonymous substitution rate (Ks) of protein-coding genes. While synonymous codons produce the same amino acids (synonymous changes), nonsynonymous changes result in different amino acids to be translated. Therefore, the Ka/Ks ratio is used to determine whether there is a selective pressure acting on the protein-coding gene; we found that the values of Ka/Ks for the GA signal transduction-related genes in both ‘*duli*’ pear and Arabidopsis were all less than 1 (Appendix A). This clearly demonstrates that these genes of ‘*duli*’ pear have been subjected to purification selection during the evolution process; hence, the individual gene pairs identified above may have similar functions between Arabidopsis and ‘*duli*’ pear.

### 2.5. Three Types of cis-Acting Elements Were Present in the Promotor Regions of PbGID1s, PbDELLAs, and PbSLYs 

Spatial and temporal expression of a gene is very important for its proper biological function. This is determined by the *cis*-acting elements in its promoter region. As shown in Figure 5 and Appendix A, three types of *cis*-acing elements were present in the promoter regions of PbGID1s: growth and development regulatory-responsive, hormone-responsive, and stress responsive elements, named here as class I, II, and III, respectively. While the types and numbers of *cis*-elements in each type were very similar, the light-responsive element GATA-motif, the 6K protein-binding site Unnamed-1, and the palisade tissue mesophyll cell differentiation element HD-zip1 in class I, as well as the TC-rich repeats in class III, were unique to *PbGID1c-1-1* and *PbGID1c-1-2*. *PbGID1c-2* was also very different in that the zeatin metabolism regulatory element O2-site and the auxin response element TGA-element in class II were not present. Interestingly, the promoter of *PbGID1c-2* contained unique *cis*-acting elements, the dehydration, low temperature, and salt stress response elements DRE1 and WRE3, and the wound and pathogen response element W box. For the promoter sequences of *PbGID1b-1* and *PbGID1b-2*, there were also some unique elements found, such as (1) the regulatory element A-box of class I present in both, (2) the F-box and the rhythm response element circadian present in class I, the GA response element GARE-motif present in class II, and the drought-induced response element MBS in class III unique to *PbGID1b-2*, and (3) the light-responsive element AT1-motif, AE-box, and ACE in class I and the salicylic acid-induced *cis*-acting element TCA-element in class II unique to *PbGID1b-1*. 

The common *cis*-acting elements found in the promoter regions of all six *PbDELLAs* were the G-Box, ABRE, ARE, and W box although the numbers were slightly different (Figure 5b). However, there were some unique elements in each of the pro*PbDELLAs*. For example, the AT1-motif of class I and TGA-element of class II were found in the promoter region of *PbGAI1b*, while the p-Box was found in *PbGAI2a*. The GCN4-motif and TATC-box were unique to pro*PbGAI2b*, the 3-AF1 binding site and the *cis*-acting element DRE were unique to pro*PbRGLa*, the TCCC-motif and O2-site were unique to *PbRGLb*, and chs-CMA2a and WRE3 were unique to pro*PbRGLa* and pro*PbRGLb*. 

The *cis*-acting elements found in the promoter regions of all *PbSLYs* were Box 4, GT1-motif, and as-1 of class I and ABRE and TGACG-motif of class II, although their numbers were slightly different (Figure 5c). The TCT-motif and CAT-box in class I, GARE-motif in class II, and MBS in class III were shared by *PbSLY1-1* and *PbSLY1-2*, while chs-CMA2a and RY-element in class I were present on both *PbSLY2-1* and *PbSLY2-2*. Unique elements, such as Gap-box, I-box, AT1-motif, F-box and transcription factor-binding site AP-1 in class I, as well as LTR and WRE3 in class III, were only found in the promoter region of *PbSLY1-1*, while chs-CMA1a and GCN4-motif of class I and TGA-element of class II were only in that of *PbSLY2-1*. Furthermore, the TATC-box, p-box, and TCA-element in class II and the WUN-motif in class III were unique to *PbSLY1-2*, while the HD-zip1 of class I and TC-rich repeats of class III were unique to *PbSLY2-2*. 

Taken together, it was found that the common *cis*-acting elements were present in the promoter regions of the genes in the same family. However, there were also unique ones found in some genes within the same family. This suggests that, while genes in the same family play similar roles, each gene also has its unique function in growth, development, hormone response, and stress response.

### 2.6. PbGID1s, PbDELLAs, and PbSLYs Were Ubiquitously Expressed with Tissue-Specific High-Level Expression for Some Genes in Each Family

In order to understand the expression profiles of *PbGID1s*, *PbDELLAs*, and *PbSLYs*, real-time RT-PCR was carried out using total RNA isolated from roots, shoots, leaves, flowers, and young fruits of 5 year old trees. The results showed that the transcripts of all candidate genes were present in all the tissues tested; however, their expression levels were different (Figure 6). For *PbGID1s*, with the exception of *PbGID1c-1-2* which was expressed in all tissues at similar levels, other *PbGID1s* were expressed at very low levels in the roots and flowers compared to in other tissues. The highest expression levels of *PbGID1c-1-1/1c-2* (Type A) were found in leaves and fruits while those of *PbGID1b-1/1b-2* (Type B) were found in new shoots. However, *PbDELLAs* showed opposite trends to *PbGID1s*, whereby their expression levels were much higher in roots and flowers, but lower in new shoots. *PbGAI1a, PbGAI1b*, and *PbGAI2b* were also expressed at high levels in leaves, while *PbGAI2b* and *RGLb* were expressed at low levels in young fruits. The main expression site for *PbSLYs* was in the reproductive tissues. The expression levels of *PbSLYs* in flowers were similar to those of *PbDELLAs*, but opposite to *PbGID1s*.

### 2.7. Changes in Expression Levels of PbGID1s, PbDELLAs, and PbSLYs in Tissue-Cultured Seedlings Treated with GA_3_, PAC, IAA, ABA, and NaCl

To see if/how the expression profiles of *PbGID1s*, *PbDELLAs*, and *PbSLYs* changed in response to phytohormones and salt stress, tissue-cultured seedlings were treated with GA_3_, PAC, IAA, ABA, and NaCl for different lengths of time. The expression levels of *PbGID1s, PbDELLAs*, and *PbSLYs* were monitored by qRT-PCR. As shown in Figure 7, in GA_3_-treated seedlings, the highest expression levels of *PbGID1c-1-1*, *1c-1-2*, and *1c-2* were reached at 12, 120, and 12 h, with 1.8-, 4.2-, and 3.6-fold increases compared to 0 h, respectively. The expression levels of *PbDELLAs* and *PbSLYs* were increased first and then decreased, with *PbGAI2a/b* decreasing to the lowest levels at 3 h after treatment. Similarly, the expression levels of all *PbSLYs* were also the lowest at 3 h, with *PbSLY2-1* and *2-2* being the most affected by GA_3_.

When the seedlings were treated with PAC, an inhibitor of GA, all *PbGID1s*, with the exception of *PbGID1c-1-2*, were downregulated first and then upregulated, with *PbGID1c-1-1* and *PbGID1c-2* being the most sensitive and reaching the lowest expression levels with 5.3-fold and 6.8-fold decreases at 6h and 3h compared that at 0 h, respectively. On the contrary, *PbDELLAs* and *PbSLYs* showed opposite expression patterns to *PbGID1s*, i.e., their expression levels increased first and then decreased in PAC-treated seedlings. The highest expression levels of *PbGAI2a* and *PbRGLb* were reached at 3 h while those of all four *PbSLYs* were increased at 72 h, with *PbSLY2-1* and *2-2* showing the most profound change with increases by 6.3- and 33.6-fold, respectively. 

In IAA-treated seedlings the expression levels of all *PbGID1s* were increased except for *PbGID1b-1*. *PbGID1c-1-1* and *PbGID1b-2* were quickly affected, with the highest expression levels with 1.3- and 2.3-fold increases achieved after 3 h, whilst *PbGID1c-1-2* was the most affected with a 3.3-fold increase after 120 h. The expression levels of all six *PbDELLAs* were decreased to start with, before increasing, with *PbGAI1a, 1b,* and *RGLb* being the most sensitive to IAA treatment with the lowest expression levels with 10.4-, 1.8-, and 5.5-fold decreases shown after 6 h. All *PbSLYs* were upregulated when treated with IAA, with *PbSLY1-1/1-2* responding the fastest and reaching the highest expression levels at 6–12 h, while *PbSLY2-1* and *2-2* were the most affected, with their expression levels increased by 7.4- and 28.1-fold, respectively.

To see if the expression of *PbGID1s*, *PbDELLAs*, and *PbSLYs* was affected by stress, ABA and NaCl were supplemented in the media. The effects of ABA and NaCl on the expression levels of these genes were very similar. In the case of *PbGID1s*, the initial downregulation of *PbGID1b**-1* and *PbGID1b-2* was the most significant with 4.2- and 6.0-fold (ABA) and 7.1- and 3.8-fold (Nacl) decreases. However, *PbDELLAs* showed opposite trends, whereby their expression levels first increased and then decreased. This was particularly true for *PbGAI2b* and *PbRGLa/b*. All *PbSLYs*, except for *PbSLY1-2* whose transcript level was increased throughout the treatment period, showed an increase then decrease. 

Therefore, these combined data indicate that GA signaling plays an important role in the phytohormonal and abiotic stress response during ‘*duli*’ pear seedling development.

## 3. Discussion

The *‘**d**uli’* pear is a wild pear species native to China, which is widely distributed in the northern region of the country. While the fruits are small and not edible, its rather well-developed root system, vigorous growth, high tolerance to biotic and abiotic stress, and good compatibility with European and Asian pear trees in grafting make it an ideal rootstock. In particular, a dwarfed rootstock is very desirable in order to achieve high density, uniformed planting, and ease of pesticide spraying and harvest.

Previous studies in wheat, rice, and other crop plants clearly showed that the plant hormone GA plays important roles in regulating plant height and stature [18,21,22,23]. As such, many GA insensitive dwarf mutants were isolated and used in agriculture to achieve high yield. Therefore, studying the GA signal transduction-related genes, expression patterns, and functions in ‘*duli*’ pear could provide valuable information for the molecular breeding of dwarf rootstocks.

### 3.1. The Genome of ‘duli’ Pear Encodes More Members of Each Family of Genes Related to GA Signal Transduction Pathway Than Other Plant Species

A total of 15 GA signal transduction-related genes were identified from the ‘*duli*’ pear genome (Figure 1, Table 1). Compared to Arabidopsis, rice, and many other plants, the number of genes in each family was increased. This seems to be correlated to a genome-wide doubling event [40]. AtGID1s in Arabidopsis have 13 domains, TWVLIS, LDR, FFHGGSF, HS, IYD, YRR, DGW, GDSSGGNI, GNI, MF, LDGKYF, WYW, and GFY, that are responsible for binding to GA or DELLA [41]. Similarly, our analysis of PbGID1s also identified TWVLIS, FFHGGSF, YRR, GDSSGNI, LDGKYF, and GFY motifs in their sequences (motifs 3, 7, 2, 5, 1, and 4, respectively, Appendix A), indicating that PbGID1s are most likely typical GID1s, and that these important functional motifs are highly conserved among plant GID1s. It is noteworthy, however, that PbGID1c-1-1 and PbGID1c-1-2 shared the highest similarity of ~94%; the only differences between them were as follows: (1) PbGID1c-1-1 had 20 more amino acids than PbGID1c-1-2 at the N-terminus; (2) only three amino acids were different within the remainder of the sequences. Both genes were also located on the same (12th) chromosome (Figure 1). Therefore, they are most likely the same gene that has been misannotated. Further cloning and sequencing will confirm this conclusion.

Compared to five in Arabidopsis, only one in rice, and one in tomato, the ‘*duli*’ pear genome contained six PbDELLAs. Further analysis showed that they belonged to three groups with two in each group, which is very similar to the DELLAs in the closely related apple genome [38]. It is worth noting that the predicted protein sequence of PbGAI1a had two DELLA sequences, which were also longer than others (Figure 3). We were puzzled by this, as this has not been previously reported in the literature. We, therefore, carried out RT-PCR, cloning, and sequencing of *PbGAI1a*. The results showed that the deduced protein sequence contained only one DELLA, not two DELLAs (data not shown). Therefore, the two DELLA domains of PbGAI1a in the database were most likely caused by error when the ‘*duli*’ pear genome sequence was assembled.

The genome of the ‘*duli*’ pear encodes four SLYs compared to two in Arabidopsis and one in rice, respectively [28,31]. All four PbSLYs contained the F-box conserved domains and the specific motifs 1, 2, and 3 (F-box, LSL, and GGF, respectively), which were shown to be essential for the function of SLY1/GID2 in Arabidopsis [30] and rice [28]. Therefore, the presence of the same motifs in PbSLYs suggests that they are also important for their function in pear. Motif 5 was only found in PbSLY1-1/2 (SLY-I) while motif 4 was found in PbSLY2-1/2 (SLY-II). The study of OsGID2/SLYI in rice showed that it contains a unique VR1 motif in its N-terminus, which is not shared with AtSLY1. However, this motif is not required for its function because both WT-OsGID2 and OsGID2-ΔVR1 can rescue the phenotype of *gid2* [28]. Therefore, whether the two groups of *PbSLYs* function differently in ‘*duli*’ pear is worth investigating in the future. 

The combined results clearly demonstrated that the ‘*duli*’ pear genome encodes all three families of GA-related signaling proteins. Despite their small variations in some specific motifs/domains were found compared to their homologous proteins in other plant species, they all contained the conserved domains and motifs important for their proper function in GA signaling.

### 3.2. Putative Functions of GA Signal-Related Genes in ‘duli’ Pear

In general, GAs are actively synthesized in young developing tissues, such as new shoots and small fruits, while well-developed mature tissues contain low levels of GAs. In line with this, we found that *PbGID1s* were expressed at low levels in roots and flowers but high levels in new shoots and young fruits, while *PbDELLAs* and *PbSLYs* (except for roots) showed an opposite trend (Figure 6). This indicates that GA can promote the expression of *PbGID1s* but inhibit that of *PbDELLAs* and *PbSLYs*. Similarly, the highest expression level of rice *OsGID2/SLY1* was found in unopened flowers, where the highest level of active GAs was detected [28]. Different expression levels of *PbGID1s*, *PbDELLAs*, and *PbSLYs* were found when ‘*duli*’ pear seedlings were treated with IAA, GA_3_, and PAC, as well as ABA and NaCl (Figure 7). The Type A *PbGID1s* were upregulated by IAA and GA3 but downregulated by PAC, indicating that they function in the growth and development of ‘*duli*’ pear seedlings. Similar results were found in tomato plants, where the three GID1s were expressed in all tissues but only GID1a responded to GA treatment, confirming the important regulatory role of GID1a in seed germination, stem elongation, and leaf development [16]. Studies in Arabidopsis and peach showed that the Arabidopsis *gid1a/gid1c* double mutant and peach single mutant *ppgid1c* were all severely dwarfed, indicating that GID1a and/or GID1c play important roles in regulating stem elongation and, hence, plant height [13,17]. These results, together with the results from our homology analysis of these genes (Figure 2), suggest that the Type A PbGID1s are most likely involved in the regulation of vegetative growth and plant stature in ‘*duli*’ pear. On the other hand, the Type B PbGID1s were downregulated in seedlings treated with ABA and NaCl; thus, they may play critical roles in abiotic stress. In supporting this, Illouz-Eliaz et al. found that tomato GID1b, a Type B GID1 maintained its stability when tomato grew in uncontrollable and unfavorable field conditions [16].

Different DELLA proteins have different roles. For example, in Arabidopsis, AtGAI and AtRGA play a major role in vegetative growth, AtRGA, AtRGL1, and AtRGL2 play a major role in seed germination and flower development, and AtRGL3 plays a major role in stress response [26,27]. The three *PbDELLAs*, *PbGAI1a*, *P**b**GAI1b*, and *PbGAI2b*, were highly expressed in roots and leaves but lowly expressed in new shoots, while *PbGAI2a*, *PbGAI2b*, *PbRGLa*, and *PbRGLb* were highly expressed in flowers but lowly expressed in young fruits except *PbRGLa* (Figure 6). Treatment with GA_3_ resulted in the downregulation of *PbGAI2a* and *PbGAI2b* (3 h), while PAC caused the upregulation of *PbGAI2a* and *PbRGLb* (3 h) (Figure 7). PbDELLAs may also play important roles in stress response. This is because, when seedlings were treated with stress hormone ABA and salt, *PbRGLa* and *PbRGLb* showed the fastest and most increased expression levels, followed by *PbGAI2a* and *PbGAI2b*, while *PbGAI1a* and *PbGAI1b* remained unchanged (Figure 7). Therefore, these expression data, combined with the results from homology and evolutionary analysis (Figure 2), indicate that *PbGAI2a, PbGAI2b, PbRGLa*, and *PbRGLb* function in the regulation of reproduction, as well as in stress, while *PbGAI1a, 1b* and *2a* play important roles in stem elongation, as well as leaf development, in ‘*duli*’ pear. It is worth noting that *PbGAI2a* and *PbGAI2b* were the only pair of genes without a syntenic relationship (Figure 2 and Figure 4a), indicating that there may be some functional differentiation of these two genes.

The expression profiles of *PbSLYs* in IAA-, GA3-, PAC-, ABA-, and NaCl-treated tissue-cultured seedlings showed similar trends to those of *PbDELLAs* but opposite trends to those of *PbGID1s*. Most interestingly, the type II PbSLYs (*PbSLY2-1* and *PbSLY2-2*) showed the most significant changes, while type I PbSLYs (*PbSLY1-1* and *PbSLY1-2*) showed the fastest changes following different treatments (Figure 7). *AtSLYs* in Arabidopsis also showed different functions because the *sly1sne* double mutant a showed more severe dwarf phenotype and significantly reduced fertility compared to the single mutant *sly1* [32]. Furthermore, overexpression of *SNE* (*SLY2*) can partially restore the GA-insensitive phenotype of *sly1* mutants [42,43]. Therefore, *SNE,* a type II SLY, can replace *SLY1*, i.e., SLY2 functions redundantly with SLY1. However, overexpression of *SLY1* in the *sly1* mutant can lead to the degradation of AtRGL1 and AtRGL2, while excessive *SNE**/SLY2* results in no change in RGL2 and only a reduction in RGL1 levels. This indicates that the functions of SNE/SLY2 and SLY1 are not exactly the same [31]. Whether the two types of *PbSLYs* function in a similar fashion to those in Arabidopsis remains to be explored.

In summary, we identified 15 GA-related signaling genes from the wildtype ‘*duli*’ pear genome. Further bioinformatic analysis and expression studies of these genes showed that they are typical and comparable to those characterized from Arabidopsis and other plant species. The type A PbGID1s play a major role in the regulation of growth and development, while type B PbGID1s play a major role in stress. The members of type I and II *PbDELLAs* (*PbGAI1a, PbGAI1b, PbGAI2a*, and *PbGAI2b*) may be involved in stem elongation and leaf (vegetative) development, while type Ⅱ and Ⅲ *PbDELLAs* (*PbGAI2a, PbGAI2b, PbRGLa*, and *PbRGLb*), as well as all four *PbSLYs*, may be involved in reproduction and stress response (Appendix A). As such, these preliminary data could guide further research into GA signaling pathways and the identification of desirable genes for ‘*duli*’ pear rootstock development.

## 4. Materials and Methods

### 4.1. Plant Materials

Roots, shoots, leaves, flowers, and young fruits of ‘*duli*’ pear (Pyrus betulifolia Bunge) were sampled from three trees (5 years old) maintained in the ‘*duli*’ Resource Orchard of Hebei Agricultural University, Hebei, China. The stem segments (about 1–2 cm) of the same trees were removed and used as explants to generate tissue-cultured seedlings for hormonal and abiotic stress treatments, as described below.

### 4.2. Identification and Chromosome Location of the Candidate GA Signaling-Related Genes

To identify the genes in the GA signaling pathway in ‘*duli*’ pear, the known amino-acid sequences of Arabidopsis and/or rice GID1, DELLA, and SLY were used as references to search the whole genome sequence of ‘*duli*’ pear (GWHA,m>AYT00000000, the Genome Warehouse of the BIG Data Center, Beijing Institute of Genomics (BIG), Chinese Academy of Sciences) [44,45]. This was followed by sequence analysis to identify and verify the Pfam and conserved domains via the CDD (https://www.ncbi.nlm.nih.gov/cdd/, accessed on 6 December 2021) [46] and SMART (http://smart.embl.de/smart/batch.pl, accessed on 7 December 2021) databases, as well as conserved motifs via MEME (http://meme-suite.org/tools/meme, accessed on 7 December 2021) [47]. The molecular weight (MW) and isoelectric point (pI) of each of the encoded proteins were predicted using the online software ExPASy (https://www.expasy.org/, accessed on 6 February 2022). The location of these genes on the specific chromosomes was mapped and marked using the software TBtools [45].

### 4.3. Evolutionary Analysis of the Putative GA Signaling-Related Genes

The amino-acid sequences of the above identified proteins in ‘*duli*’ pear were compared with those from other plants using Clustal (https://www.ebi.ac.uk/Tools/msa/clustalo/, accessed on 7 February 2022). A phylogenetic tree was constructed using the maximum likelihood method by applying the JTT matrix model and the MEGA6.0 software with default parameters and1000 bootstraps. This was imported to the online tool iTOLs (https://itol.embl.de/itol.cgi, accessed on 7 February 2022) to produce the figure.

### 4.4. Synteny and Gene Duplication Analysis

The syntenic relationship of the genes with a role in the GA signaling between Arabidopsis and pear was analyzed. The orthologous and paralogous relationships between them were drawn using TBtools [45]. The Ka, Ks, and Ka/Ks values for the individual gene pairs were calculated by the adjoint method using the software KaKs_Calculator 2.0 (( https://sourceforge.net/projects/kakscalculator2/, accessed on (8 March 2022) [48].

### 4.5. Prediction and Analysis of cis-Elements in the Promoter Regions of the GA Signaling Genes of ‘duli’ Pear

In order to identify relevant *cis*-acting elements, the DNA sequences about 2000 bp upstream of the transcriptional start codon ATG of the *PbGID1, PbDELLA,* and *PbSLY* genes of ‘*duli*’ pear were analyzed using the PlantCARE online database (http://bioinformatics. psb.ugent.be/webtools/plantcare/html/, accessed on 12 December 2021) [49].

### 4.6. Treatment of Tissue-Cultured ‘duli’ Pear Seedlings with Gibberellin, Paclobutrazol, Auxin, Abscisic Acid, and Salt

To assess the response of the candidate genes to phytohormones and stress, GA_3_ (2 mg/L), PAC (2 mg/L), IAA (0.2 mg/L), ABA (4 mg/L), and NaCl (0.6%) were individually added to MS medium used to culture the ‘*duli*’ pear seedlings. After 0, 3, 6, 12, 72, and 120 h, the seedlings were collected and frozen immediately in liquid nitrogen and stored at −80 °C for further qRT-PCR analysis.

### 4.7. Total RNA Extraction and First-Strand cDNA Synthesis

The total RNA was isolated from roots, young shoots, leaves, flowers, fruits, and tissue-cultured ‘*duli*’ seedlings of pear plant using the RNAprep Pure Plant Kit according to the manufacture’s protocol (TIANGEN, Beijing, China). The first-strand cDNA was synthesized from 1 µg of total RNA using the First-Strand cDNA Synthesis Kit (TIANGEN, Beijing, China).

### 4.8. qRT-PCR Assay

The transcript levels of the identified genes were analyzed by qRT-PCR using the above-synthesized cDNAs as templates, 2×*TransStart*^®^ Top Green qPCR SuperMix (TransGen Biotech, Beijing, China), and gene-specific primers (Appendix A) in a 20 μL reaction mix. The reaction was run on a Step-Two Plus real-time PCR system (Applied Biosystems, San Francisco, CA, USA). *PbACTIN* was used as the reference gene. At least three replicates were included in each experiment, and experiments were repeated three times.

## 5. Conclusions

Bioinformatic analysis showed that there were five GA receptors (PbGID1s), six GA negative regulators (PbDELLAs), and four GA positive regulators (PbSLYs) in the ‘*duli*’ pear genome. The expression profiles of these genes showed that, although they were ubiquitously expressed, their expression levels and trends were different. Exogenous GA and IAA application could increase the transcript levels of *PbGID1s* but decrease those of *PbDELLAs* and *PbSLYs*, while PAC, ABA, and NaCl showed the opposite effect on the expression of these genes. Future research should focus on the detailed functional analysis of these genes so that their involvement in plant height, reproduction, and stress response can be revealed. 

## Figures and Tables

**Figure 1 ijms-23-06570-f001:**
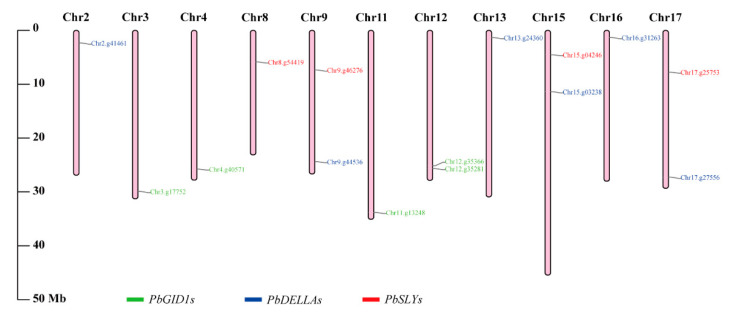
Chromosomal locations of *PbGID1s*, *PbDELLAs*, and *PbSLYs*. Pink vertical bars denote the chromosomes of ‘*duli*’ pear. The scale on the left indicates the chromosome lengths (Mb). The different families of GA signal genes are color-coded: *PbGID1s*, green; *PbDELLAs*, blue; *PbSLYs*, red.

**Figure 2 ijms-23-06570-f002:**
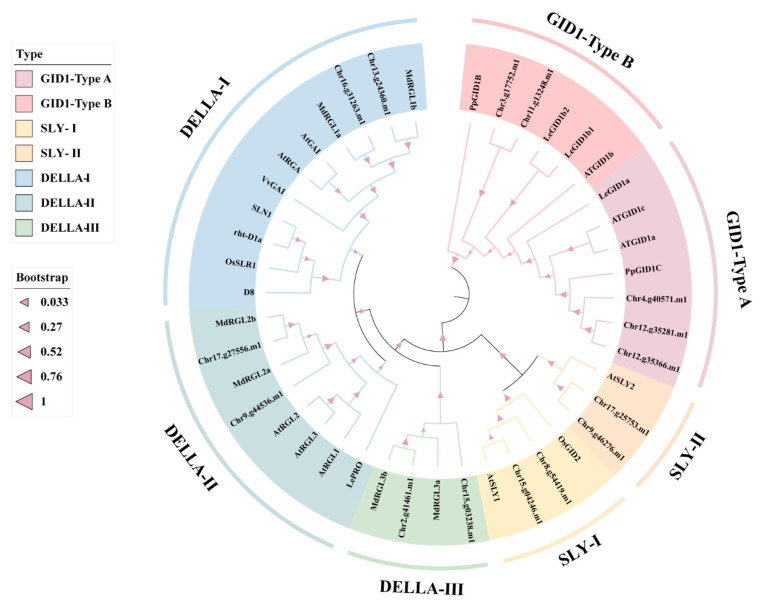
Phylogenetic tree of PbGID1s, PbDELLAs, and PbSLYs of ‘*duli*’ pear (*Pyrus betulifolia* Bunge). Multiple sequence alignment was performed using the encoded protein sequences of the candidate genes with Clustal software. MEGA 6.0 was applied to construct the maximum likelihood (ML) tree using the JTT matrix model with 1000 bootstrap replicates. The different groups of PbGID1s, PbDELLAs, and PbSLYs are color-coded as indicated on the top left panel.

**Figure 3 ijms-23-06570-f003:**
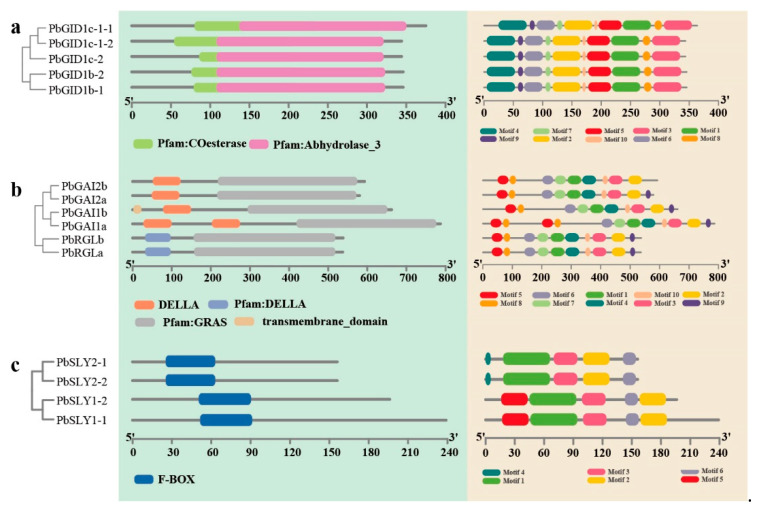
Identification of the conserved domains and motifs of PbGID1s, PbDELLAs, and PbSLYs in ‘*duli*’ pear (*Pyrus betulifolia* Bunge). (**a**) The conserved domains and motifs of PbGID1s. (**b**) The conserved domains and motifs of PbDELLAs. (**c**) The conserved domains and motifs of PbSLYs. Individual domains (in green background on the left) and motifs (in peach background on the right) are indicated in different colored horizontal bars below each figure. Numbers on the *X*-axis indicate amino-acid positions of the proteins.

**Figure 4 ijms-23-06570-f004:**
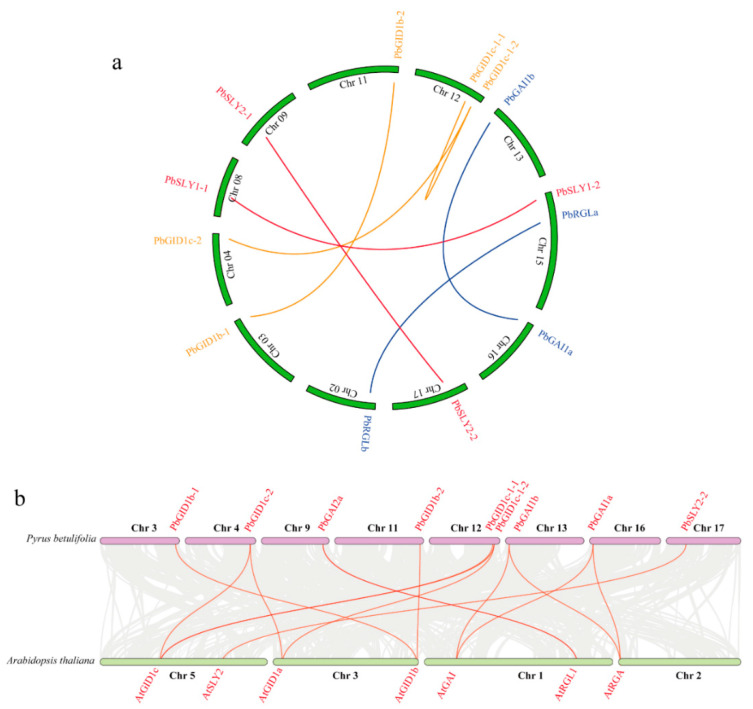
Gene duplication of *PbGID1s*, *PbDELLAs*, and *PbSLYs* of ‘*duli*’ pear and their synteny with homologous genes in Arabidopsis. (**a**) Gene duplication of *PbGID1s* (orange), *PbDELLAs* (blue), and *PbSLYs* (red) within the ‘*duli*’ pear genome. Duplicated genes are linked by lines of the same color in each family. (**b**) The syntenic relationship of gene pairs, linked by a red line, between ‘*duli*’ pear and Arabidopsis.

**Figure 5 ijms-23-06570-f005:**
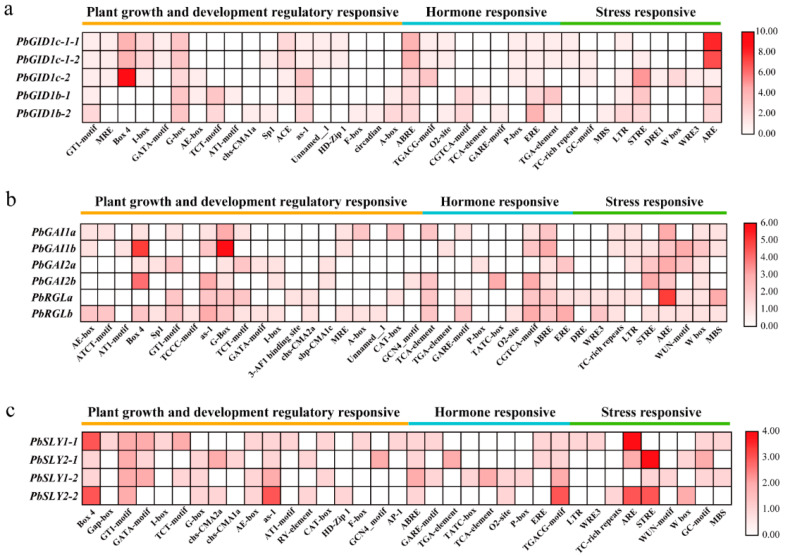
The *cis*-acting elements identified in the promoter regions of *PbGID1s*, *PbDELLAs*, and *PbSLYs* of ‘*duli*’ pear (*Pyrus betulifolia* Bunge). (**a**) The *cis*-acting elements in the promoter region of *PbGID1s*. (**b**) The *cis*-acting elements in the promoter region of *PbDELLAs*. (**c**) The *cis*-acting elements in the promoter region of *PbSLYs*. The types (bottom of the charts) and numbers (bars on the right) of *cis*-acting elements in the 2000 bp upstream of the transcriptional start codon are shown in the heatmap.

**Figure 6 ijms-23-06570-f006:**
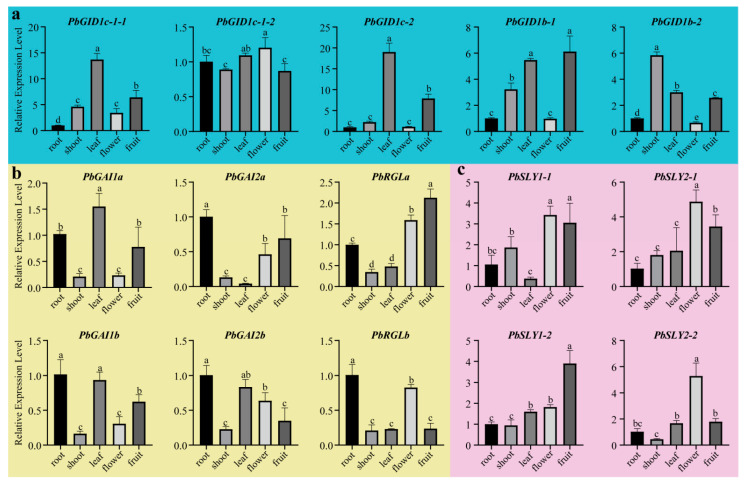
Expression profiles of *PbGID1*, *PbDELLA*, and *PbSLY* family genes in different tissues of ‘*duli*’ pear (*Pyrus betulifolia* Bunge). (**a**) Expression profiles of *PbGID1s* (shaded blue). (**b**) Expression profiles of *PbDELLAs* (shaded yellow). (**c**) Expression profiles of *PbSLYs* (shaded lilac). Quantitative RT-PCR was carried out using total RNA isolated from different tissues as indicated. *PbACTIN* was used as the reference gene. The relative expression level of each gene was calculated relative to the transcript level of the roots (set as 1). Each bar represents the mean ± SE (*n* = 3). Different letters above the bars indicate significant differences at *p* < 0.05 (*n* = 3) analyzed by Duncan’s multiple range test.

**Figure 7 ijms-23-06570-f007:**
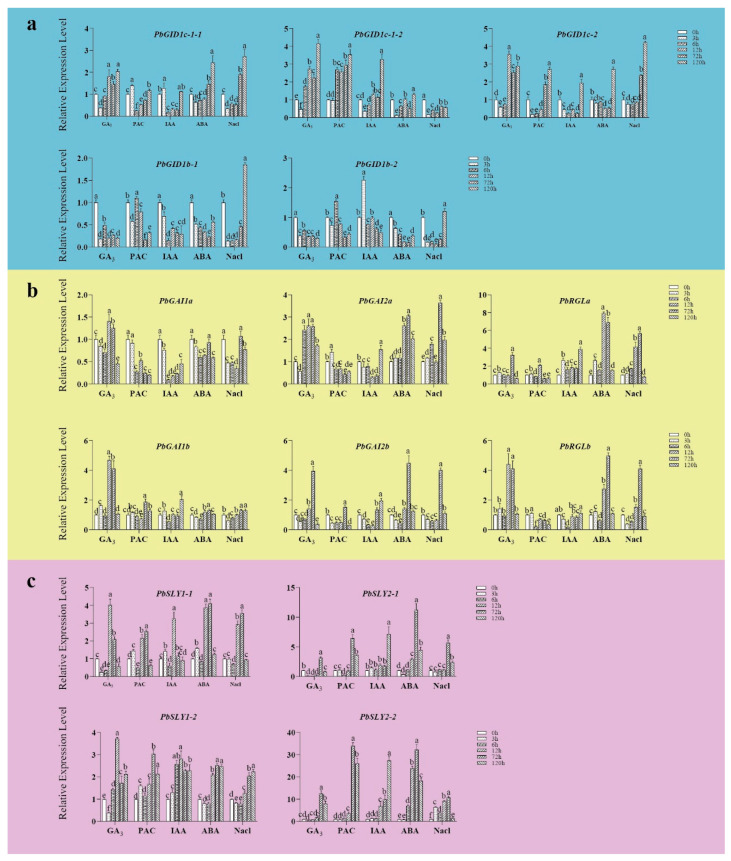
Expression profiles monitored by real-time RT-PCR of *PbGID1s*, *PbDELLAs*, and *PbSLYs* in ‘*duli*’ tissue-cultured seedlings treated with GA_3_, PAC, IAA, ABA, and NaCl. (**a**) Expression profiles of *PbGID1s* (shaded blue). (**b**) Expression profiles of *PbDELLAs* (shaded yellow). (**c**) Expression profiles of *PbSLYs* (shaded lilac). Quantitative RT-PCR was carried out using total RNA isolated from tissue-cultured ‘*duli*’ seedlings treated with GA_3_ (2mg/L), PAC (2 mg/L), IAA (0.2 mg/L), ABA (4 mg/L), and NaCl (0.6%) for 0, 3, 6, 12, 72, and 120 h. *PbACTIN* was used as the reference gene. The relative expression level of each gene was calculated relative to the transcript level of that gene at 0 h (set as 1). Each bar represents the mean ± SE (*n* = 3). Different letters above the bars indicate significant differences at *p* < 0.05 (*n* = 3) analyzed by Duncan’s multiple range test.

**Table 1 ijms-23-06570-t001:** Features of *PbGID1, PbDELLA*, and *PbSLY* genes identified from the genome of ‘*duli*’ pear (*Pyrus betulifolia* Bunge).

Gene Name	Gene ID	Chr.	Genomic Location	CDS	Exon	AA	MW (kDa)	pI
*PbGID1c-1-1*	Chr12.g35366.m1	12	Chr12(+)25128010-25129710	1095	2	364	41.23	8.54
*PbGID1c-1-2*	Chr12.g35281.m1	12	Chr12(-)25667020-25668660	1035	2	344	38.80	7.56
*PbGID1c-2*	Chr4.g40571.m1	4	Chr4(-)25772601-25774240	1035	2	344	38.60	6.51
*PbGID1b-2*	Chr11.g13248.m1	11	Chr11(-)33811347-33812718	1041	2	346	39.25	7.90
*PbGID1b-1*	Chr3.g17752.m1	3	Chr3(-)29907597-29908997	1041	2	346	39.18	7.09
*PbGAI2b*	Chr17.g27556.m1	17	Chr17 (+)27245475-27249911	1782	2	593	64.89	4.81
*PbGAI2a*	Chr9.g44536.m1	9	Chr9 (+)24381403-24383145	1782	1	593	63.00	5.01
*PbGAI1b*	Chr13.g24360.m1	13	Chr13 (+)1332016-1334706	1989	2	662	72.84	5.53
*PbGAI1a*	Chr16.g31263.m1	16	Chr16 (+)1311639-1316924	2364	2	787	86.32	4.87
*PbRGLb*	Chr2.g41461.m1	2	Chr2 (+)2328831-2331799	1617	2	538	59.04	5.43
*PbRGLa*	Chr15.g03238.m1	15	Chr15 (+)11383910-11385523	1614	1	537	58.79	5.54
*PbSLY2-1*	Chr9.g46276.m1	9	Chr9(-)7349856-7350326	471	1	156	17.50	8.50
*PbSLY2-2*	Chr17.g25753.m1	17	Chr17(-)7751252-7751722	471	1	156	17.44	8.38
*PbSLY1-2*	Chr15.g04246.m1	15	Chr15(+)4519363-4519953	591	1	196	21.60	7.98
*PbSLY1-1*	Chr8.g54419.m1	8	Chr8(+)5866651-5867630	720	2	239	26.26	6.25

## Data Availability

Not applicable.

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
