# Peer review of "Genome-Wide Analysis of Genes Involved in the GA Signal Transduction Pathway in ‘*duli*’ Pear (*Pyrus betulifolia* Bunge)"

_ijms, 2022, doi:10.3390/ijms23126570_

Round 1

Reviewer 1 Report

The presented manuscript describes the analysis of genes related to GA signaling in the "duli" pear. The work is well structured and written, and after a few corrections it will be suitable for publication.

Section 2.2 - please make it clear that it is the description of proteins (for which the phylogenetic tree was created), not genes.

Line 132 - the colour used is blue, not “purple”.

Lines 208 and 477- please swap "&" with "and"

Line 248 - the word "Segmental" should be in lowercase

Line 462 - a dot is missing after "et al"

Figures 5 and 7: the font should be larger. Especially the Fig. 7 is illegible, the bars should be wider, and the letters above the bars are not visible at all (if not for the caption, I would not even notice them).

The spaces in front of the square brackets are often missing throughout the work.

Having corrected these minor shortcomings, I think that the manuscript should be published due to its high merit value.

Reviewer 2 Report

Very intersting work. Only few changes are required:

Objectievs of the work must be clarified in a separated parragraph without referbnces.

A new section of conlcuisons must be added.

Plant material assyaed must be clarified in the Material and Methods section.

Reviewer 3 Report

Manuscript ID: ijms-1705357

In this ms ID: ijms-1705357, entitled “Genome-wide Analysis of Genes Involved in the GA Signal Transduction Pathway in ‘duli’ Pear (Pyrus betulifolia Bunge)”, the Authors reported and characterized in detail the genes involved in GA signalling in pear ‘duli’ by a combined approach of in silico analysis of the recently sequenced genome of Pyrus betulifolia Bunge and Real Time qPCR experiments. Their work is very interesting, because they provided evidence of the involvement of 15 genes in GA signalling in ‘duli’ pear. Significantly, these data pave the way to further studies on molecular mechanisms of GA signalling in pear.

However, in several points results were not clearly described so the paper has to be revised. English language, typos, and repetitions have also to be checked before publication. This reviewer recommends important revisions of this paper.

Here are some specific comments:

ABSTRACT

L18: remove repetition of “pear”

L23: explain new acronyms

INTRO

L32-33: use plural and introduce acronym GA: “Gibberellin or gibberellic acids (GA)…”; use plural changing “It” to “They” and so on.

L34: add “response” after “stress”

L35: check “then” before “rice”

L42: add SLY

L60: check English in “play overlap yet …”

L74: fix typo in “-Iie-”

L108: fix typo in “DELIA”

RESULTS

L186: remove repetition of “based”

L193, 205: add Figure 3a and 3b

L233: in general, here add the concept of “dispersed” duplication

L253: remove “can”

L257: use “have been” instead of “are”, check “purify”

Par. 2.4: Collinearity is when sets of homologous genes in different species are located on the same chromosome (synteny) and are conserved in the same order. In light of this, the description in L229-333 (Fig. 4a) and L234-241 (Fig. 4b) have to be opportunely changed.

L274: check unicity of cis-elements, HD-zip1 is not the only unique element: GATA-motif etc.

L275-276: it also lacks the GATA-motif

L284: add AT1-motif

L285: use plural “elements”

L290: add 3AF1

L291: add CMA2a

L307: check “reproduction”, should use “hormone response”

Par. 2.6: L317-327 and Figure 6. Here, description of results of relative gene expression have to be opportunely changed, considering a threshold and relative gene expression of 1 indicated no variation. Please, see work of Livack and Schmittgen (2001) available at https://doi.org/10.1006/meth.2001.1262, they reported that the amount of target, normalized to an endogenous reference and relative to a calibrator, is given by the formula: 2−ΔΔCt. Eventually, by using log2(fold change), values are symmetrical and can see fold changes in both directions (+/-). As an important consequence, in your results a significant expression exist only for some tissues, while gene expression level in root is identical (equal to 1) for GIDs, DELLAs, and SLYs.

Par. 2.7: L338-374 and Figure 7. Here, as above, description of results of relative gene expression have to be opportunely changed, considering a threshold and relative gene expression of 1 indicated no variation. Above all in L362-363, they are general trends with some exceptions, this reviewer suggests avoiding net divisions.

L336,376,448: avoid using the term “Altered”, they are only relative physiological responses.

DISCUSSION

L392: add bibliographic reference(s)

L393: check verb tense for “are”

L402: avoid repetitions “duplication doubling”

L412: mis-sequenced ? Please, provide evidence

L416: remove comma

L421: remove “human”

L437: remove the first “important”

L441: use plural for GA

L433: revise expression levels in root and flowers (L320-327)

L450: revise expression levels

L453: remove “was”

L458: change “with” to “of”

L465-480: DELLAs levels are low in roots, revise after Results

L481-482: after revising Results, here revise expression levels

L490-491: rewrite this sentence more clearly

M&M

Add a new paragraph with “Plant material” and introduce the Latin name Pyrus betulifolia Bunge of wild pear

L523: remove repetition of “based”

L500: use “assess” instead of “see”

Conclusions

You have to conclude with main findings and perspectives.

L558: remove this sentence, it is already in the results

L560: write about the applications of the other factors in addition to GA

Round 2

Reviewer 2 Report

Authoirs have revised correctly the manuscript